# Asymmetric Within-Document Predictive Learning for Scientific Document Representation

You Zuo[1,2]    Éric de la Clergerie[2]    Benoît Sagot[2]

(1) Questel, Paris, France    (2) Inria, Paris, France

`firstname.lastname@inria.fr`

## ABSTRACT

We study predictive pretraining for scientific document representation using the discourse structure of papers. We propose SciJEPA, a citation-free framework that learns through asymmetric within-document prediction: title and abstract representations are used to predict method representations, and method representations are used to predict conclusion representations. Experiments on RELISH, high-influence citation, SciDocs, and cite prediction show that plain predictive training is viable but weaker than a controlled contrastive baseline using the same section pairs. Adding Sliced Isotropic Gaussian Regularization (SIGReg) substantially improves performance and narrows this gap. The effect of regularization is task-dependent: moderate SIGReg helps fine-grained ranking, while stronger regularization can weaken local alignment. We further show that different encoding branches support different retrieval regimes. These results position within-document predictive learning as a promising citation-free complement for scientific document representation, provided that embedding geometry is carefully controlled.

## RÉSUMÉ

**Apprentissage prédictif asymétrique intra-document pour la représentation de documents scientifiques**

Nous étudions le pré-entraînement prédictif pour la représentation de documents scientifiques en exploitant la structure discursive des articles. Nous proposons SciJEPA, un cadre sans supervision par citations qui apprend par prédiction asymétrique intra-document : les représentations du titre et du résumé sont utilisées pour prédire les représentations de la section méthodologique, puis les représentations de la section méthodologique pour prédire celles de la conclusion. Des expériences sur RELISH, high-influence citation, SciDocs et cite prediction montrent que l'apprentissage prédictif seul est viable, mais reste inférieur à une baseline contrastive contrôlée utilisant les mêmes paires de sections. L'ajout de la régularisation gaussienne isotrope esquissée (SIGReg) améliore nettement les performances et réduit cet écart. L'effet de la régularisation dépend de la tâche : une SIGReg modérée aide le classement fin, tandis qu'une régularisation plus forte peut affaiblir l'alignement local. Nous montrons également que différentes branches d'encodage soutiennent différents régimes de recherche. Ces résultats positionnent l'apprentissage prédictif intra-document comme un complément prometteur, sans supervision par citations, pour la représentation de documents scientifiques, à condition que la géométrie des plongements soit soigneusement contrôlée.

KEYWORDS: self-supervised learning, Joint Embedding Predictive Architecture (JEPA), document representation.

MOTS-CLÉS : apprentissage auto-supervisé, architecture prédictive à plongements conjoints (JEPA), représentation de documents.

# 1 Introduction

Learning effective representations of scientific documents is central to scholarly retrieval, citation recommendation, literature discovery, and document ranking. Strong scientific document encoders such as SPECTER (Cohan *et al.*, 2020) and SPECTER2 (Singh *et al.*, 2023) typically encode the title and abstract of a paper and learn from cross-document supervision derived from citation links or related scholarly tasks. This strategy is highly effective, especially on citation-oriented benchmarks, but citation links define a particular supervision signal: they are delayed for newly published papers (Xing *et al.*, 2021), vary across fields and article types (Radicchi *et al.*, 2008), and serve rhetorical functions beyond semantic similarity, such as providing background, acknowledging methods or datasets, supporting claims, or contrasting prior work (Jurgens *et al.*, 2018). As a result, citation-supervised encoders combine textual semantics with graph-induced scholarly relatedness. In this work, we ask whether useful scientific document representations can be learned from a paper's content and internal discourse structure alone, without citation links, and whether these citation-free representations transfer to citation-related retrieval tasks. We view this signal as complementary to citation supervision rather than as a replacement when dense citation graphs are available.

To study this question, we propose SciJEPA, a Joint-Embedding Predictive Architecture for scientific document representation. Scientific papers follow a structured progression from problem statement to methodology and findings or implications: the title and abstract describe the problem and contribution, the method section specifies how the problem is addressed, and the conclusion summarizes outcomes and implications. A useful content-based representation should support this inference: from a high-level summary, one should infer the likely methodology, and from the method, plausible conclusions or outcomes.

SciJEPA operationalizes this idea with two asymmetric within-document prediction tasks: title and abstract → method, and method → conclusion. Following Joint-Embedding Predictive Architectures (LeCun *et al.*, 2022), SciJEPA predicts the target section in representation space rather than reconstructing it token by token. This is important because earlier sections do not determine the exact wording or details of later sections, but they do constrain their broader discourse role and semantic content. The objective therefore encourages the model to learn dependencies between problem statements, methods, and conclusions without citation links or negative examples during pretraining.

We evaluate SciJEPA on scientific retrieval benchmarks from SciRepEval (Singh *et al.*, 2023) and compare it with a controlled contrastive baseline using the same corpus, backbone, and section pairs. Plain predictive training learns useful citation-free representations but remains weaker than contrastive learning; adding SIGReg (Balestriero & LeCun, 2025) substantially improves performance and narrows this gap. Our analysis identifies embedding geometry as a key bottleneck: without regularization, predictive training tends to produce anisotropic representations whose variance is concentrated in a few dominant directions. These findings indicate that within-document prediction is a viable citation-free signal when embedding geometry is carefully controlled.

# 2 Related Work

**Scientific document representation.** Scientific document encoders range from domain-adapted language models such as SciBERT (Beltagy *et al.*, 2019) to document-level retrieval models trained

with cross-document supervision. Building on SciBERT as its foundation, SPECTER (Cohan *et al.*, 2020) learns embeddings of titles and abstracts using citation-based triplets, and SPECTER2 (Singh *et al.*, 2023) extends this approach through multi-format training with adapters. These models are strong on citation-derived benchmarks because their supervision is closely aligned with the evaluation signal. Recent work such as SemCSE (Brinner & Zarriess, 2025) explores non-citation semantic supervision through LLM-generated summaries and contrastive positive pairs. SciJEPA instead isolates a citation-free and non-contrastive signal from the internal discourse structure of individual papers.

**Document structure and facets.** Prior work also exploits scientific document structure. Hierarchical architectures such as HDT (He *et al.*, 2024) improve long-document modeling through hierarchical attention. CoSAEmb (Singh & Singh, 2024) uses full-text sections as aspect-specific views and trains section-aware embeddings with supervised contrastive triplet loss. FLeW (Dou *et al.*, 2025) uses citation intent and citation frequency to learn background-, method-, and result-oriented representations. These methods show that sections and facets are useful, but typically use structure for long-document encoding, contrastive matching, or citation-informed facet learning. SciJEPA uses structure differently: section roles define asymmetric prediction tasks within the same paper, where title–abstract, method, and conclusion are treated as discourse components connected by directional dependencies rather than interchangeable views.

**Predictive learning and geometry.** Joint-Embedding Predictive Architectures (JEPA) were proposed as an alternative to reconstruction-based and contrastive self-supervised learning (LeCun *et al.*, 2022). Instead of reconstructing inputs or contrasting positive and negative pairs, JEPA predicts the latent representation of a target view from a context view. This paradigm has been successful in image and video learning (Assran *et al.*, 2023; Bardes *et al.*, 2023, 2024) and has recently been explored for language (Huang *et al.*, 2025; Gillin *et al.*, 2026). Because non-contrastive predictive methods lack explicit negatives, they raise questions about collapse and anisotropy. LeJEPA (Balestriero & LeCun, 2025) studies scalable non-contrastive joint-embedding learning and introduces Sliced Isotropic Gaussian Regularization (SIGReg) as a way to encourage non-collapsed and more isotropic representations. We build on this regularization idea and study how SIGReg affects global isotropy, dominant-component concentration, and local retrieval alignment in scientific document embeddings.

# 3 Methodology

## 3.1 Asymmetric Within-document Prediction

We formulate scientific document pretraining as an *asymmetric within-document prediction* problem. Given a context section $x_c$ and a target section $x_t$, the model encodes each section into a latent representation and learns to predict the target representation from the context representation. For scientific papers, we instantiate this framework with two directional tasks:

$$\text{title+abstract} \rightarrow \text{method}, \tag{1}$$

$$\text{method} \rightarrow \text{conclusion}. \tag{2}$$

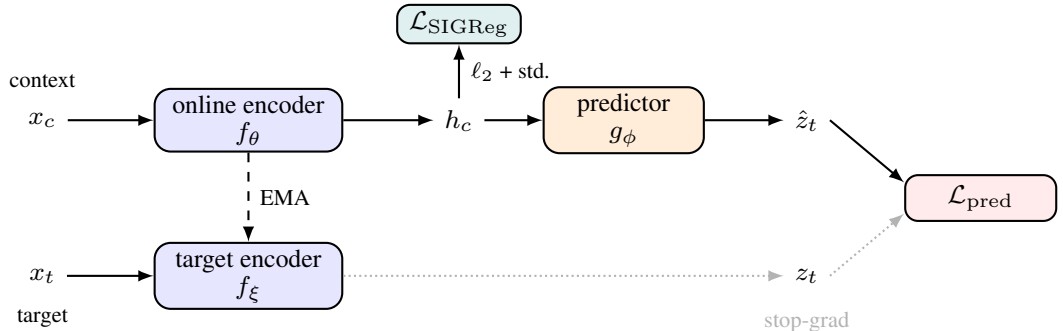

FIGURE 1 – Overview of SciJEPA for title+abstract $\rightarrow$ method and method $\rightarrow$ conclusion. The online encoder predicts a target representation through a predictor, while the target encoder is EMA-updated and stop-gradient. SIGReg regularizes the normalized online representation before the predictor; preprocessing details are given in Appendix B.

The first task predicts methodological content from a summary-level context, while the second predicts outcome-oriented content from methodological information. These pairs are not interchangeable views: their direction reflects discourse dependencies within the paper.

## 3.2 Predictive Architecture and Regularization

Figure 1 summarizes the SciJEPA architecture. In non-contrastive latent prediction, the loss only enforces agreement between learned representations. Without negatives or token reconstruction, a symmetric agreement objective may collapse by mapping different inputs to identical or nearly identical embeddings. SciJEPA therefore follows the online–target design used in non-contrastive self-supervised learning (Grill *et al.*, 2020; Chen & He, 2021; LeCun *et al.*, 2022; Assran *et al.*, 2023): the online branch is optimized by back-propagation, while the target branch provides a slowly evolving reference representation.

Given a context section $x_c$ and a target section $x_t$, the online encoder $f_\theta$ produces a pooled context representation $h_c = f_\theta(x_c)$, while the target encoder $f_\xi$ produces $z_t = f_\xi(x_t)$. Gradients are stopped on $z_t$, so the target representation is fixed for the current update. The target encoder is not directly optimized by the predictive loss; instead, after each update, its parameters are updated as an exponential moving average (EMA) of the online encoder parameters:

$$\xi \leftarrow m\xi + (1 - m)\theta, \tag{3}$$

where $m \in [0, 1)$ is the EMA momentum, set close to 1 so that the target encoder changes slowly. Thus, the target branch follows the learned representation space with a delay, providing a stable reference for prediction and reducing the risk of representational collapse.

The predictor $g_\phi$ maps the context representation into the target latent space through a residual transformation:

$$\hat{z}_t = h_c + g_\phi(h_c). \tag{4}$$

This predictor is needed because the context and target sections play different discourse roles. A title–abstract representation should be related to a method representation, but it should not be forced

to be identical to it. The predictor therefore absorbs part of the transformation between section roles while preserving the online encoder output as a general document representation.

For each training example $i$, the predictive loss matches the predicted representation $\hat{z}_{t,i}$ to the stop-gradient target representation $z_{t,i}$:

$$\ell_{\text{pred}}^{(i)} = 1 - \cos(\hat{z}_{t,i}, z_{t,i}). \tag{5}$$

The batch predictive loss is $\mathcal{L}_{\text{pred}} = \frac{1}{B} \sum_i \ell_{\text{pred}}^{(i)}$.

The online–target design stabilizes the prediction target, but it does not control how document embeddings are distributed globally. Since $\mathcal{L}_{\text{pred}}$ contains no negatives, different documents are not explicitly encouraged to occupy distinct regions of the space. We add SIGReg (Balestriero & LeCun, 2025) to the batch of normalized online representations before the predictor. SIGReg regularizes random one-dimensional projections of the batch distribution toward a Gaussian reference, discouraging collapse and dominant-direction concentration.

The final training objective is

$$\mathcal{L} = \frac{1}{B} \sum_{i=1}^{B} \ell_{\text{pred}}^{(i)} + \lambda \mathcal{L}_{\text{SIGReg}}. \tag{6}$$

By default, evaluation uses the online encoder output $f_\theta(x)$, which serves as the directly optimized general document representation. For analysis, we also compare the predictor branch $f_\theta(x) + g_\phi(f_\theta(x))$ and the target branch $f_\xi(x)$.

## 3.3 Controlled Contrastive Baseline

A key question is how much performance can be obtained from the same citation-free within-document section pairs under a strong non-citation baseline. Since section-based contrastive learning has been effective for technical-document representation learning (Zuo *et al.*, 2025), we include a controlled CL baseline with the same encoder backbone, pretraining corpus, section extraction procedure, and positive section pairs as SciJEPA. Unlike SciJEPA, CL replaces latent prediction with InfoNCE and in-batch negatives. This baseline does not isolate the effect of negatives, but it provides a strong comparison under the same section-pair supervision.

Given two views $x_i^{(1)}$ and $x_i^{(2)}$ from the same document, the encoder produces pooled and $\ell_2$-normalized representations $h_i^{(1)}$ and $h_i^{(2)}$. We optimize an InfoNCE loss with in-batch negatives from other documents:

$$\mathcal{L}_{\text{cl}}^{(i)} = -\log \frac{\exp\left(\text{sim}(h_i^{(1)}, h_i^{(2)})/\tau\right)}{\exp\left(\text{sim}(h_i^{(1)}, h_i^{(2)})/\tau\right) + \sum_{j \neq i} \exp\left(\text{sim}(h_i^{(1)}, h_j^{(1)})/\tau\right)}, \tag{7}$$

where $\text{sim}(\cdot, \cdot)$ is cosine similarity and $\tau$ is the temperature. The final contrastive loss averages Equation (7) over the batch. This baseline is not intended to reproduce citation-supervised SPECTER-style training.

# 4 Experimental Setup

## 4.1 Pretraining Data

We pretrain on S2ORC-ArXiv [1], a Hugging Face release of S2ORC (Lo *et al.*, 2020), which provides structured full text for open-access papers. We filter papers with very short abstracts, abnormal section counts, short bodies, or malformed headings, and remove papers whose `corpus_id` appears in any evaluation benchmark. After filtering and decontamination, we obtain 1.14M unique retained documents. During training, we sample section-pair examples from this set with replacement until reaching a fixed budget of approximately 5M training examples. Thus, 5M refers to the number of sampled training examples rather than the number of unique documents.

For each paper, we extract title+abstract, method, and conclusion/results sections using heading-keyword matching with positional fallbacks. Papers without method sections are discarded; papers without conclusions contribute only the title+abstract $\rightarrow$ method task. Additional extraction details and statistics are provided in Appendix A.

## 4.2 Model and Training Details

Unless otherwise stated, all models use SciBERT (Beltagy *et al.*, 2019) as the encoder backbone and mean pooling over non-padding tokens. The predictor is a two-layer MLP with architecture of $\text{LayerNorm}(d) \rightarrow \text{Linear}(d, d_p) \rightarrow \text{GELU} \rightarrow \text{Linear}(d_p, d)$, where $d = 768$ and $d_p = 2048$. Its output is added residually to the encoder representation, as in Equation (4). The final linear layer is zero-initialized, so the predictor starts close to the identity map and gradually learns a target-oriented transformation. The target encoder uses the EMA schedule:

$$m_t = m_{\text{final}} - (m_{\text{final}} - m_{\text{base}}) \cdot \tfrac{1}{2}(1 + \cos(\pi t/T)), \tag{8}$$

with $m_{\text{base}} = 0.996$ and $m_{\text{final}} = 0.999$. When enabled, SIGReg is applied to the online encoder representation before the predictor using 512 random projection slices; implementation details are described in Appendix B.

We train with AdamW, weight decay 0.01, encoder learning rate $5 \times 10^{-6}$, predictor learning rate $1 \times 10^{-4}$, cosine decay, 50 warmup steps, and gradient clipping at 1.0. We define one training epoch as one pass over a fixed stream of 5M section-pair examples sampled with replacement from the pool of 1.14M unique retained documents. All models are trained for one such epoch on a single NVIDIA H100 GPU with per-device batch size 512 and gradient accumulation over 8 micro-batches (effective batch size 4,096). We use gradient checkpointing and `bf16` mixed precision.

## 4.3 Baselines

We consider two groups of baselines. First, for a controlled objective-level comparison, we train models with the same encoder backbone, pretraining corpus, section extraction procedure, and optimization pipeline: **CL** (*contrastive learning*), **SciJEPA**, and **SciJEPA + SIGReg**. The CL model replaces the predictive loss with an InfoNCE objective. Positive pairs are constructed from the

---

1. https://huggingface.co/datasets/AlgorithmicResearchGroup/s2orc_arxiv

same section pairs used by SciJEPA, and negatives are drawn from other documents in the same in-batch pool. Since training uses a per-device batch size of 512 with 8 gradient accumulation steps, the InfoNCE negatives are computed within each 512-example micro-batch, while the effective optimization batch size remains 4,096 for all controlled models. The temperature follows a cosine ramp from 0.05 to 0.1 after warmup. This baseline is intended to isolate the effect of the training objective, not to reproduce citation-supervised SPECTER-style training.

Second, we compare against publicly available scientific-document encoders used off the shelf: **SciBERT** with mean pooling (Beltagy *et al.*, 2019), **SPECTER** (Cohan *et al.*, 2020), and **SPEC-TER2** (Singh *et al.*, 2023). [2] For SPECTER2, we use the benchmark-specific adapter recommended by the SciRepEval setup.

## 4.4   Evaluation

| Benchmark | Format | Relevance signal | Candidate regime |
|---|---|---|---|
| RELISH | Ranking | Expert similarity | Curated semantic candidates |
| High-influence | Ranking | Key-citation score | Citation-related candidates |
| SciDocs-Cite | Ranking | Direct citation | Task-specific candidates |
| SciDocs-CoCite | Ranking | Shared citations | Task-specific candidates |
| SciDocs-CoView | Ranking | Co-view behavior | Task-specific candidates |
| SciDocs-CoRead | Ranking | Co-read behavior | Task-specific candidates |
| Cite prediction | Triplet | Citation-positive pair | Positive–negative pair |

TABLE 1 – Evaluation protocols and relevance signals. RELISH and high-influence citation require ranking among curated or citation-related candidates and are relatively fine-grained, whereas cite prediction is a coarser triplet discrimination task. SciDocs tasks rank task-specific candidate sets using citation-, co-citation-, co-view-, or co-read-based relevance.

We evaluate on scientific retrieval benchmarks that differ in both relevance signal and candidate granularity: RELISH (Brown & Zhou, 2019), high-influence citation (Singh *et al.*, 2023), SciDocs (Cohan *et al.*, 2020) (*cite*, *co-cite*, *co-view*, and *co-read*), and cite prediction (Cohan *et al.*, 2020). We report NDCG for RELISH, MAP for high-influence citation, MAP and NDCG for SciDocs, and triplet accuracy for cite prediction. Unless otherwise stated, we use SPECTER and SPECTER2 embeddings without $\ell_2$-normalization, following their original setup. For SciBERT and our trained models, we $\ell_2$-normalize embeddings before evaluation. Table 1 summarizes the evaluation protocols.

These protocol differences matter for interpretation. First, several benchmarks are directly citation-aligned, so citation-supervised encoders such as SPECTER and SPECTER2 are evaluated under signals close to their training objective. SciJEPA, by contrast, never observes citation links during pretraining; these tasks therefore test whether within-document discourse prediction transfers to cross-document retrieval. Second, candidate granularity differs across tasks: RELISH and high-influence citation require fine-grained ranking among relatively related candidates, while cite prediction tests coarser positive–negative discrimination. This helps interpret if models show different performance patterns across benchmarks.

---

2. We use the public Hugging Face checkpoints: https://huggingface.co/allenai/scibert_scivocab_uncased, https://huggingface.co/allenai/specter, and https://huggingface.co/allenai/specter2.

**Geometry diagnostics.** In addition to retrieval metrics, we analyze embedding geometry using alignment and uniformity (Wang & Isola, 2020), and singular spectrum deviation (SSD), using the dimension-normalized definition of Zuo *et al.* (2025), motivated by prior analyses of anisotropy through singular-value spectra (Godey *et al.*, 2024). Alignment measures the distance between related document pairs; lower values indicate that positives are closer. Uniformity measures how evenly embeddings spread over the hypersphere; lower values indicate better global spread. SSD measures deviation of the dimension-normalized singular-value spectrum from an isotropic reference; lower values indicate less dimensional concentration or anisotropy. These diagnostics are not retrieval objectives, but help characterize how the embedding space balances local matching and global geometric regularity.

We compute these diagnostics on a fixed sample of 2,000 cite-prediction triplets. Uniformity and SSD are computed over all 6,000 query, positive, and negative embeddings, while alignment is computed over the 2,000 query–positive pairs.

# 5 Results

## 5.1 Main Results

| Model | RELISH NDCG | High-Influence MAP | Cite-Pred. Acc. |
|---|---|---|---|
| SciBERT | 85.17 | 38.58 | 81.88 |
| SPECTER | 90.07 | 42.88 | **94.31** |
| SPECTER2 + Adapters | **91.86** | **46.07** | 93.95 |
| CL | 90.25 | 43.76 | 92.12 |
| SciJEPA | 87.46 | 40.58 | 90.25 |
| SciJEPA + 0.001· SIGReg | 89.14 | 43.39 | 91.37 |
| SciJEPA + 0.0025· SIGReg | 89.71 | 44.06 | 91.15 |
| SciJEPA + 0.005· SIGReg | 89.93 | 43.98 | 90.40 |

TABLE 2 – Main results on RELISH, high-influence citation, and cite prediction.

| Model | SciDocs-Cite MAP / NDCG | SciDocs-CoCite MAP / NDCG | SciDocs-CoView MAP / NDCG | SciDocs-CoRead MAP / NDCG |
|---|---|---|---|---|
| SciBERT | 65.97 / 82.77 | 71.00 / 85.93 | 69.96 / 84.48 | 67.12 / 82.89 |
| SPECTER | **92.29** / 96.73 | 88.19 / 94.83 | 83.64 / 91.54 | 85.34 / 92.87 |
| SPECTER2 + Adapters | 92.23 / **96.83** | **91.14** / **96.28** | **85.21** / **92.27** | **86.86** / **93.53** |
| CL | 87.20 / 94.34 | 89.86 / 95.65 | 84.03 / 91.61 | 86.12 / 93.23 |
| SciJEPA | 81.53 / 91.37 | 85.64 / 93.55 | 81.28 / 90.28 | 80.79 / 90.40 |
| SciJEPA + 0.001· SIGReg | 85.98 / 93.78 | 88.30 / 94.86 | 83.09 / 91.23 | 84.28 / 92.29 |
| SciJEPA + 0.0025· SIGReg | 85.84 / 93.72 | 87.92 / 94.72 | 82.84 / 91.12 | 84.15 / 92.17 |
| SciJEPA + 0.005· SIGReg | 84.49 / 93.08 | 87.00 / 94.33 | 82.06 / 90.75 | 83.55 / 91.92 |

TABLE 3 – Main results on the four SciDocs retrieval tasks.

Tables 2 and 3 compare SciJEPA with controlled and off-the-shelf baselines. Plain SciJEPA improves substantially over the SciBERT backbone on all benchmarks, showing that asymmetric within-document prediction provides a useful citation-free training signal. However, it remains below the controlled CL baseline in most tasks, indicating that predictive training is currently less robust than contrastive learning with in-batch negatives under the same section-pair supervision.

Adding SIGReg improves SciJEPA across all tasks and substantially narrows the gap to CL. The strongest result appears on high-influence citation, where SciJEPA + 0.0025· SIGReg reaches 44.06 MAP, slightly outperforming CL and ranking close to SPECTER2 + Adapters. This suggests that geometric regularization is a central ingredient for making predictive within-document learning competitive.

Compared with off-the-shelf scientific encoders, SciJEPA remains competitive but does not surpass the strongest citation-supervised baselines. SPECTER2 with task-specific adapters achieves the best results on most tasks, while SPECTER remains strongest on cite prediction and SciDocs-Cite MAP. This is expected given the citation alignment of many evaluation benchmarks [3] and reinforces our main interpretation: within-document prediction is a viable citation-free signal, but citation-supervised encoders retain an advantage on citation-derived tasks.

The optimal SIGReg weight is task-dependent: stronger regularization helps RELISH and high-influence citation, whereas smaller values work better for cite prediction and SciDocs. We analyze this trade-off in the next section.

## 5.2 Effect of SIGReg

To study geometric regularization, we vary the SIGReg weight $\lambda$ and evaluate both downstream performance and the geometry diagnostics defined in Section 4.4: alignment, uniformity, and SSD (Figure 2). The effect of SIGReg is non-monotonic and task-dependent: RELISH peaks at $\lambda = 0.005$, high-influence citation at $\lambda = 0.0025$, while cite prediction and SciDocs favor smaller regularization.

A plausible explanation is that these tasks differ in candidate granularity. RELISH and high-influence citation require fine-grained ranking among curated or citation-related candidates. In this setting, reducing dominant-direction effects can help: if unregularized SciJEPA concentrates variance in a few global directions, subtle differences among already related papers may be compressed. Moderate SIGReg spreads variance more evenly across dimensions and can improve ranking resolution.

The geometry diagnostics support this view. As $\lambda$ increases, SSD decreases, indicating less concentration in a few singular directions. However, stronger SIGReg tends to weaken local alignment beyond the moderate range. This helps explain why cite prediction and SciDocs, which rely on keeping citation- or usage-related positives close to separate them from broader candidate sets, prefer smaller $\lambda$. One interpretation is that SIGReg increases the effective dimensionality of the embedding space: this helps when dominant directions compress fine-grained distinctions, but excessive regularization may spread variance into weakly task-relevant directions. In distance-based retrieval, these extra directions can act as noise, increasing pairwise distances and weakening positive-pair alignment. The best $\lambda$ therefore depends on whether a benchmark benefits more from fine-grained ranking resolution or from compact local neighborhoods.

---

3. Prior work (Ostendorff *et al.*, 2022) reports metadata overlap between SPECTER's training corpus and SciDocs evaluation papers, though not with SciDocs gold labels.

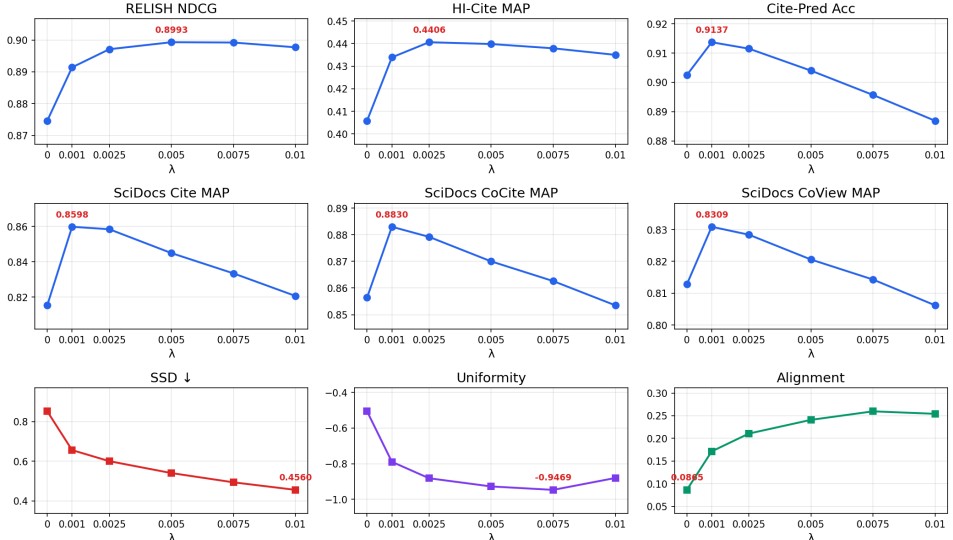

FIGURE 2 – Effect of the SIGReg weight $\lambda$ on downstream performance and embedding geometry. Moderate regularization improves several retrieval benchmarks, especially RELISH and high-influence citation. Larger $\lambda$ generally reduces spectral concentration but can weaken alignment and hurt some task metrics, revealing a trade-off between global geometric regularity and local similarity preservation.

## 5.3 Analysis of Encoding Branches

SciJEPA exposes three possible inference representations: the online encoder, the predictor output, and the EMA target encoder. All our main results use the online encoder by default, but Figure 3 shows that the branches behave differently across tasks. On fine-grained ranking benchmarks such as RELISH and high-influence citation, the online encoder performs best. By contrast, on cite prediction and average SciDocs retrieval, the predictor output outperforms the online encoder across a broad range of $\lambda$, while the target encoder remains consistently weakest.

This suggests that the predictor is not merely a disposable training head. The online encoder appears to preserve fine-grained local distinctions, which helps when candidates are already related and must be carefully ranked. The predictor output, however, applies a target-oriented transformation learned for section prediction; this may smooth or reweight the representation in a way that is less sensitive to stronger SIGReg and more suitable for coarser citation-style discrimination. The target encoder is useful as a stable EMA training target, but its lagged parameters make it less effective as an inference representation.

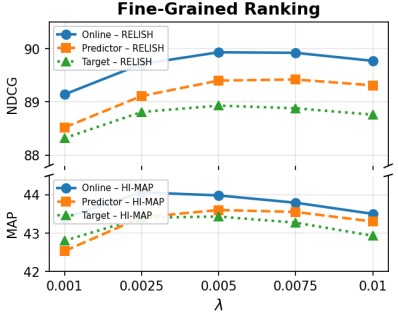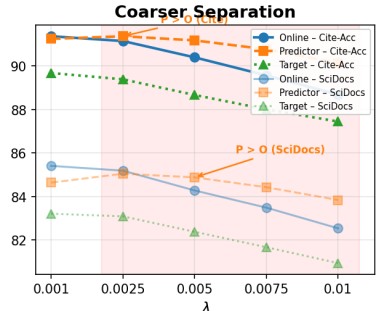

FIGURE 3 – Comparison of online encoder, predictor, and target encoder representations across SIGReg weights. The online encoder performs best on fine-grained ranking tasks, while the predictor output is more robust on cite prediction and average SciDocs retrieval. The target encoder remains consistently weaker.

# 6 Conclusion

We introduced SciJEPA, a citation-free predictive pretraining framework for scientific document representation based on asymmetric within-document section prediction. SciJEPA learns useful representations without citation links, but plain predictive training remains weaker than a controlled contrastive baseline using the same section-pair supervision.

Our results show that embedding geometry is central to making predictive learning effective: SIGReg improves SciJEPA, but its effect is task-dependent and overly strong regularization can weaken local alignment. Branch analysis further suggests that different parts of the predictive architecture support different retrieval regimes. Overall, within-document predictive learning is a promising citation-free complement to citation-supervised scientific document representation.

# Acknowledgments

We thank the anonymous reviewers for their insightful comments, the CLEPS infrastructure at Inria Paris for computational resources, and Younes Djemmal, Kim Gerdes, and Kirian Guiller for their helpful discussions and feedback.

This work was partly funded by the last author's chair in the PRAIRIE institute funded by the French national agency ANR as part of the "Investissements d'avenir" programme under the reference ANR-19-P3IA-0001.

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

# A    Section Extraction Details

We extract discourse segments from the structured section text provided by S2ORC-ArXiv. Each paper is represented as a sequence of section titles and section bodies. We construct three segments: title+abstract, method, and conclusion/results.

**Filtering.**    To reduce parsing noise, we discard papers with abstracts shorter than 50 characters, fewer than 3 or more than 80 sections, body text shorter than 1,000 characters, or more than 30% malformed section titles. A section title is considered malformed by a simple heuristic if it satisfies any of the following: (i) title length $> 120$ characters, (ii) more than four periods, or (iii) it contains sentence-like phrases such as `if and only if`, `we have that`, `it follows`, or `this holds`. These patterns capture PDF parsing artifacts where running text is mistakenly parsed as a heading. We also remove all papers whose `corpus_id` appears in any evaluation benchmark.

**Method section.**    We identify method sections by matching section headings against a regex that covers method-related terms (e.g., `method`, `methodology`, `approach`, `framework`, `model architecture`, `technical approach`, `problem formulation`). If no heading match is found, we use two fallbacks: (i) select the first substantial section (content length $\geq 50$ characters) after an `introduction` heading; (ii) if no explicit introduction is found, select the second substantial section in the paper. Papers for which no substantial method-like section can be extracted are discarded.

**Conclusion/results section.**    We identify conclusion/result-oriented sections after the selected method section by heading match (e.g., `conclusion`, `conclusions`, `results`, `results and discussion`, `discussion`, `summary`, `findings`). If no heading match is found, we use the last substantial section after the method section as a fallback. If no such section exists, the paper is retained but only contributes the title+abstract $\rightarrow$ method prediction task.

**Task construction.**    For papers with both method and conclusion/results sections, we construct two possible prediction tasks:

$$\text{title+abstract} \rightarrow \text{method}, \qquad \text{method} \rightarrow \text{conclusion/results.}$$

During training, one of the available tasks is sampled for each example. If both tasks are available, they are sampled uniformly; otherwise, only the title+abstract $\rightarrow$ method task is used.

**Extraction statistics.**    After filtering, decontamination, and section extraction, we obtain 1.14M unique retained documents. During training, we sample with replacement from this pool until reaching a fixed budget of 5,000,000 section-pair training examples. Decontamination excludes 495,096 evaluation `corpus_ids` under the current evaluation-task set. In an automatic audit on 200,000 filtered/decontaminated streamed examples, 198,094 retained papers produced a valid method target (100% of retained by construction), and 98.50% of retained papers produced a non-empty conclusion/results target. Under the training-time sampling rule, this implies expected task proportions of 50.75% for title+abstract $\rightarrow$ method and 49.25% for method $\rightarrow$ conclusion/results.

# B    SIGReg Implementation Details

We apply Sliced Isotropic Gaussian Regularization (SIGReg) to the online encoder representation before the predictor. Let $H \in \mathbb{R}^{B \times d}$ be a batch of online representations. We first normalize each representation to unit norm,

$$\tilde{h}_i = \frac{h_i}{\|h_i\|_2},$$

and then standardize each dimension within the batch,

$$z_{ij} = \frac{\tilde{h}_{ij} - \mu_j}{\sigma_j + \epsilon},$$

where $\mu_j$ and $\sigma_j$ are the batch mean and standard deviation of dimension $j$. We denote the resulting batch by $Z \in \mathbb{R}^{B \times d}$.

We sample $K$ random unit directions $a_1, \ldots, a_K \in \mathbb{S}^{d-1}$ and compute projections

$$p_{ik} = \langle z_i, a_k \rangle.$$

For each projection direction, SIGReg compares the empirical characteristic function of $\{p_{ik}\}_{i=1}^{B}$ to the characteristic function of a standard normal distribution,

$$\varphi_{\mathcal{N}(0,1)}(t) = \exp(-t^2/2).$$

Following SIGReg (Balestriero & LeCun, 2025), we use an Epps–Pulley-style characteristic-function statistic, which provides a differentiable discrepancy between the empirical one-dimensional projection distribution and a standard Gaussian reference:

$$\mathcal{L}_{\text{SIGReg}} = \frac{1}{K} \sum_{k=1}^{K} B \sum_{\ell=1}^{L} w_\ell \left[ \left( \frac{1}{B} \sum_{i=1}^{B} \cos(t_\ell p_{ik}) - e^{-t_\ell^2/2} \right)^2 + \left( \frac{1}{B} \sum_{i=1}^{B} \sin(t_\ell p_{ik}) \right)^2 \right].$$

where $K$ is the number of random projection slices, $t_1, \ldots, t_L$ are the characteristic-function evaluation points, and $w_\ell$ are numerical integration weights. In our experiments, we use $K = 512$, $L = 17$, and uniformly spaced $t_\ell \in [0,3]$, with trapezoidal weights multiplied by the Gaussian characteristic-function weight.

**Interpretation.**    The Gaussian reference in this test is the one-dimensional standard normal distribution. Since embeddings are first $\ell_2$-normalized, SIGReg should not be interpreted as matching an unconstrained Gaussian distribution in $\mathbb{R}^d$. A useful idealized reference is the uniform distribution on the sphere: if $u \sim \text{Unif}(\mathbb{S}^{d-1})$, then for any unit vector $a$,

$$\sqrt{d} \langle a, u \rangle \xrightarrow{d} \mathcal{N}(0,1) \qquad \text{as } d \to \infty.$$

Thus, Gaussian random-projection matching after normalization is consistent with encouraging sphere-like spread in high dimension. However, our additional dimension-wise standardization is not an exact spherical-uniformity test. It acts as an empirical scale calibration for the unit-variance Gaussian reference and also equalizes coordinate-wise variances. We therefore interpret normalized

and standardized SIGReg as discouraging collapsed, highly anisotropic, or dimensionally concentrated embeddings, rather than as exactly enforcing uniformity on the sphere.

A more principled regularizer for normalized embeddings could directly match the projection distribution of the uniform sphere, or explicitly balance spherical spread with local alignment; we leave this direction for future work.