# OpenReview forum: "Asymmetric Within-Document Predictive Learning for Scientific Document Representation"
_ls2n.fr/CORIA-TALN/2026/Workshop/ARTS — ls2n CORIATALN 2026 Workshop ARTS Submission_

### Official Review · Reviewer_fSrN · 2026-04-23

**Mode De Presentation:** Oral

**Confience:**

Oui

**Decision:**

Accepté

**Relecture:**

*Disclaimers: This paper was not properly anonymized at the time of submission for review. I did not consider any author information, and this had no impact on my evaluation.*

--

This paper is an interesting proposition regarding representation of scientific articles, the methods used are modern and seem well aligned with authors intents. Current popular approaches for representing scientific such as SPECTER2 can be indeed limited due to their reliance on citation information.

As I am less familiar with JEPA architectures, this review will primarily focus on areas of the paper that I found less clear or more difficult to fully understand without expertise and that would benefit from further clarifications.

- A brief explanation of the differences between the benchmarks used would be helpful, particularly regarding how their tasks differ and how well they align with the prediction task proposed in this paper. This would improve understanding of why this specific training objective is important for article representation and for downstream applications such as recommendation or retrieval, rather than relying solely on citation-based methods, which appear to perform better across most reported metrics.

- My (limited) understanding of these datasets suggests that citation-based approaches may have an inherent advantage on these specific evaluation criteria. A more detailed explanation would therefore help clarify why SciJEPA can still be considered a promising alternative to these methods.

- Regarding geometry-related metrics, I feel like this section would benefit from further explanation on how these metrics are relevant for both controlled and off-the-shelf baselines. It is indicated that "Lower is better for all three metrics" but higher results is highlighted for alignement in SIGReg ablation study.

- Detailed JEPA architecture could benefit from more clarity on what the goal of each component is, I did not understand what are the findings related to the analysis of each branches (a term i also wasn't familiar with) and how subsequent research could improve on this.

- "all models use SciBERT (Beltagy et al., 2019) as the encoder backbone" -> as SciBERT is the less performing model on this evaluation task, does this imply that using a different encoder backbone would lead to improved performances for the JEPA architecture?

**Comments Suggestions And Typos:**
- Some acronyms are not defined on their first appearance: JEPA , EMA, CL
- Figure 1 isn't cited within the article.

**Resume:**

This article addresses the challenge of representing scientific documents by leveraging discourse structure instead of modeling the semantic space through inter-document relationships.
The authors propose using the internal discourse organization of a paper, most specifically the relational signals between sections such as the title and abstract, methods, and conclusions to predict within-document discourse relationships. The intent is to use these relationship as an alternative to contrastive learning which is difficult to model within articles collections.
Experiments are conducted using JEPA architecture, along with various regularization strategies implemented through SIGReg and how these approaches performs compared to other baselines.
Results indicate that SciJEPA is competitive compared to constrastive approaches, but does not surpass strongest citation-based baselines.

---

### Official Review · Reviewer_caCe · 2026-05-05

**Mode De Presentation:** Poster

**Confience:**

Oui

**Decision:**

Accepté

**Relecture:**

This is an interesting pilot study of an alternative architecture. In general, the paper is difficult to read.
- The paper could benefit from a more pedagogical introduction of the concepts
- The work might be better motivated
- It would be helpful to have a short description of the used baselines, measures, datasets, and SciDocs task

**Resume:**

The paper presents asymmetric within-document predictive learning for scientific document representation, focusing on the following tasks: using the title and abstract to predict the latent representation of the method section, and using the method section to predict the latent representation of the conclusion section. The authors studied whether Joint-Embedding Predictive Architectures (JEPA) can be a viable alternative to contrastive learning for scientific document representation. The experiments were conducted on RELISH, high-influence citation, SciDocs, and cite prediction datasets. The results suggest that the JEPA-style predictive objective can be a promising research direction despite its high sensitivity to embedding geometry.

---

### Decision · Program_Chairs · 2026-05-07

Accept (Poster)